# Response of Normal and Low-Phytate Genotypes of Pea (*Pisum sativum* L.) on Phosphorus Foliar Fertilization

**DOI:** 10.3390/plants10081608

**Published:** 2021-08-05

**Authors:** Petr Škarpa, Marie Školníková, Jiří Antošovský, Pavel Horký, Iva Smýkalová, Jiří Horáček, Radmila Dostálová, Zdenka Kozáková

**Affiliations:** 1Department of Agrochemistry, Soil Science, Microbiology and Plant Nutrition, Faculty of AgriScience, Mendel University in Brno, Zemědělská 1, 613 00 Brno, Czech Republic; marie.skolnikova@mendelu.cz (M.Š.); jiri.antosovsky@mendelu.cz (J.A.); 2Department of Animal Nutrition and Forage Production, Faculty of AgriScience, Mendel University in Brno, Zemědělská 1, 613 00 Brno, Czech Republic; pavel.horky@mendelu.cz; 3Agritec Plant Research Ltd., Zemědělská 2520/16, 787 01 Šumperk, Czech Republic; smykalova@agritec.cz (I.S.); horacek@agritec.cz (J.H.); dostalova@agritec.cz (R.D.); 4Institute of Physical and Applied Chemistry, Faculty of Chemistry, Brno University of Technology, Purkyňova 118, 612 00 Brno, Czech Republic; kozakova@fch.vutbr.cz

**Keywords:** chlorophyll content, fluorescence parameters, seed yield, seed quality, seed nutrient content, pea, foliar application

## Abstract

Phosphorus (P) is an important nutrient in plant nutrition. Its absorption by plants from the soil is influenced by many factors. Therefore, a foliar application of this nutrient could be utilized for the optimal nutrition state of plants. The premise of the study is that foliar application of phosphorus will increase the yield of normal-phytate (*npa*) cultivars (CDC Bronco a Cutlass) and low-phytate (*lpa*) lines (1-2347-144, 1-150-81) grown in soils with low phosphorus supply and affect seed quality depending on the ability of the pea to produce phytate. A graded application of phosphorus (H₃PO₄) in four doses: without P (P0), 27.3 mg P (P1), 54.5 mg P (P2), and 81.8 mg P/pot (P3) realized at the development stages of the 6th true leaf led to a significant increase of chlorophyll contents, and fluorescence parameters of chlorophyll expressing the CO_2_ assimilation velocity. The P fertilization increased the yield of seeds significantly, except the highest dose of phosphorus (P3) at which the yield of the *npa* cultivars was reduced. The line 1-2347-144 was the most sensible to the P application when the dose P3 increased the seed production by 42.1%. Only the *lpa* line 1-150-81 showed a decreased tendency in the phytate content at the stepped application of the P nutrition. Foliar application of phosphorus significantly increased ash material in seed, but did not tend to affect the protein and mineral content of seeds. Only the zinc content in seeds was significantly reduced by foliar application of P in *npa* and *lpa* pea genotypes. It is concluded from the present study that foliar phosphorus application could be an effective way to enhance the pea growth in P-deficient condition with a direct effect on seed yield and quality.

## 1. Introduction

Pea (*Pisum sativum* L.) is a plentifully grown leguminous plant in many countries, and it can be utilized both in human nutrition as well as a part of the feed for farm animals. It is considered as one of the most important sources in human nutrition, because its pods contain a great content of proteins, carbohydrates, vitamins, and minerals. By its ability to hold nitrogen from the air and return it back to the soil, the pea contributes to sustainable agriculture [1]. Besides high demands for nitrogen, the pea belongs to those plants with relatively high demands for phosphorus (P), too.

Crop production on more than 30% of the world arable land is limited by the P availability [2]. P is an essential nutrient required by all plants to grow, photosynthesize, and form proteins. It is especially limiting in organic environments for legumes, which need more P than cereals to form root nodules for nitrogen fixation [3,4]. In particular, P is important for the growth of the field pea, and for protein synthesis [4]. Sufficient amounts of phosphorus taken by plants are also necessary for an optimal production yield. In the case of low P supplies in the soil or conditions decreasing its intake by plants, fertilization by phosphorus significant increases not only legume production, but also its quality [5,6,7,8,9,10,11]. One of the phosphorus forms naturally presented in plants is phytic acid. Phytic acid is the major storage form of phosphorus in seeds of legumes. In the case of the pea, its highest concentration is in the endosperm [12]. Since phytate can form complexes with proteins and minerals, reducing the digestive availability of phosphorus [13], it is usually regarded as an antinutrient, although recent works indicate that it has important beneficial roles as an antioxidant for plants [14]. Therefore, there is an interest in the assessment and manipulation of phytate contents in important food grains such as peas. Through plant breeding, it is possible to prevent the phosphorus transformation into phytic acid and, thus, to decrease the content of phytic acid by even up to 50–95% [15]. For example, Wilcox et al. [16] reported an 80% decrease in P-phytate content in the low-phytate soybean. Reduction of phytate phosphorus concentration in the low-phytate pea seed by about 60% accompanied by an increase in free inorganic phosphorus was presented by Warkentin [17]. The content of phytic acid in the pea is influenced not only by various properties, but also by the climate, irrigation, and soil conditions as well as by the fertilization. The decrease of phytate concentration caused by phosphorus fertilization was observed in the case of soya [18] oat [19] and corn [20].

One possible way to provide necessary nutrients to the plants during vegetative stages is a foliar application of fertilizers. The foliar fertilization directly applies the nutrient to the plant tissue, by-passing a potential fixation, and losses that may arise from the soil application. However, the efficacy of the foliar fertilizer is relatively uncertain [21]. An uptake potential of foliar P is generally considered to be low, but the increased efficiency of the P usage has been reported with foliar application [22]. The greatest benefits of the foliar P fertilization were observed under low moisture and highly P-deficient soil conditions. In these aforementioned conditions, the foliar P application has a potential to increase the yield and quality of seeds [23,24]. However, there have been relatively few reports on analyses of the productivity and seed quality of legumes [10] or normal [11] and low-phytate pea cultivars grown under varying foliar P fertilization levels.

This work contributes to the extension of knowledge on the foliar phosphorus application and its influence on selected growth parameters, production, and quality of normal-phytate cultivars, and low-phytate pea lines. It offers an alternative approach to optimize phosphorus nutrition of peas, i.e., the use of foliar fertilization with this nutrient, which is a suitable procedure especially in conditions where soil application seems ineffective (e.g., inappropriate soil pH, which immobilizes phosphorus in the soil). A positive effect of the foliar P application on pea seed production and its quality in plants grown under P deficient conditions, especially a reduction of the portion of P bounded to phytate, is expected. An increase of free P (inorganic) for nutritional purposes is expected. At the same time, it is determined whether the low-phytate lines will use the P supplied by fertilization directly for binding to phytate, i.e., will this increase the phytate phosphorus content or will it only increase the total P content of the seeds and the phytate level will remain unchanged?

## 2. Results and Discussion

### 2.1. The Effect of Foliar P Fertilization on Content of Chlorophyll and Chlorophyll Fluorescence Parameters

The foliar application of phosphorus had a significant effect (*p* ≤ 0.05) on the content of chlorophyll in leaves of all tested pea genotypes. The increase of the N-tester values was significant (*p* ≤ 0.05, 0.01) in both measurements, especially at variant P2 and P3. A significant effect of the foliar phosphorus application on the chlorophyll content of wheat was presented by Waraich et al. [25]. Besides wheat, the increased content of chlorophyll after the phosphorous fertilization was also achieved in the case of mung bean [26] aubergine [27] maize [28] and cluster bean [29]. The positive effect of the applied phosphorus can be explained by its direct involvement into the structure of cell membranes [30] a whole range of proteins, nucleic acids, and nucleotides [25] with direct effects on photosynthesis [31]. This fact explains the decrease of the chlorophyll content by the phosphorus deficiency in rice [32]. The mean N-tester values of all tested pea genotypes measured in the first term (T1) were enhanced by the stepped P application by 2.9% (P1), 4.2% (P2), and up to 4.9% (P3). The highest increase of the N-tester values was determined in plants of the line 1-150-81 at variant P3. The effect of the phosphorus fertilization lasted up to the second term of the measurement (T2), as presented in Table 1.

The quantum yield of the electron transport of the photosystem II (*Φ_PSII_*), which expresses the real capacity of the photosystem II (PSII) for photochemical reactions, and represents the availability of reaction centers of the PSII, was significantly (*p* ≤ 0.05, 0.01) influenced by the phosphorus application. Its mean value measured at the variant without the P fertilization (P0) was 0.800. By the stepped P fertilization, this value was enhanced at all doses (P1−3) to 0.822 in the first term (T1). The results corresponded with conclusions in the study of Xu et al. [32], which found out that the phosphorus deficit had induced changes in efficiency of excitation energy absorption of reaction centers of the PSII in rice plants and had reduced the quantum yield *Φ_PSII_*. For example, the decrease of the quantum yield (*Φ_PSII_*) caused by the phosphorus deficit was observed in *Lonicera pampaninii* [33]. The photosynthetic rates in the *lpa* and *npa* genotypes of soybean were 1.3 and 1.5 times higher, respectively, in the treatment with the high P dose than with the low P dose [34]. In the first measurement term (T1), the *Φ_PSII_* value was most significantly enhanced after the phosphorus application in the case of the line 1-150-81. According to Fryer et al. [35], a strong linear correlation between quantum yield of the electron transport of the PSII (*Φ_PSII_*) and the carbon fixation efficiency was found. Plant species with phosphorus deficit like *Lotus japonicus* showed a decrease of the maximal rate of photosynthesis. In the case of the ratio of the dark respiration to the maximal photosynthesis, it declined significantly [36]. The significantly (*p* ≤ 0.05; 0.01) increased *Φ_PSII_* values of variants with the P fertilization on the *lpa* lines lasted until the later vegetation stages (T2). Contrary to this, the quantum yield values of the *npa* cultivars were reduced (Table 1). In the case of the cultivar CDC Bronco, the decrease was significant (*p* ≤ 0.01). The increase of the *Φ_PSII_* values at the *lpa* pea lines was followed by a significant enhancement of the fluorescence decrease ratio (*R_Fd_*), which is measured at saturation irradiance, and which is directly proportional to the net CO_2_ assimilation rate. In the second term T2, a significant correlation (*r* = 0.635; *p* ≤ 0.001) between the *Φ_PSII_* and *R_Fd_* values was determined at these pea lines. Therefore, it can be stated that the foliar phosphorus application has a significant effect on the availability of reaction centers on the photosystem II at the *lpa* pea lines.

### 2.2. The Effect of Foliar P Fertilization, on Yield Parameters of Pea

The seed yield of normal phytate varieties was on average 27.3% higher than the seed yield of low-phytate lines. Research comparing yield levels of *lpa* and *npa* pea genotypes indicates that the low-phytate lines were similar in agronomic performance to normal phytate cultivars, except for somewhat slower time to flowering and maturity, slightly lower seed weight, and slightly lower grain yield [17,37]. However, the primary focus of our study was to evaluate the effect of phosphorus foliar fertilization on yield and seed quality of the tested pea genotypes.

The foliar application of phosphorus significantly influenced the yield of the pea seeds (*p* ≤ 0.05; 0.01). Its enhancement was evident at all tested genotypes, as presented in Figure 1. A significantly increased grain yield was also achieved according to the literature dealing with the evaluation of the phosphorus application influence on legumes. The attention of contemporary research is mostly focused on the phosphorus uptake by the root system after the application of P-fertilizers in stepped doses to the soil. This application is used for the legumes growing both in a monoculture: pea [5], faba bean [6], chickpea [7], and mung bean [8], and in a mixed cultures [9]. Only a few literal sources evaluate the efficiency of the foliar P application on the seed production of legumes, and their quality [10,11]. Comparing the different techniques, the foliar P fertilization has a better potential to improve its nutritional deficiencies in plants caused by the low content of P in soil or limited availability of this nutrient by the root [23,38]. The reaction of the tested pea lines on the foliar P fertilization was different. In the case of the *npa* cultivars, the seed production was enhanced by the stepped doses of the P-fertilizer, except of the highest level (P3) which reduced the seed production. The yield was decreased by the mean value of 10% compared to the variant P2 at both tested cultivars. This fact corresponded to the parameters of photosynthesis determined during vegetation and indicated that the dose of 81.8 mg P (P3) is not beneficial to plants of the *npa* cultivars. Froese et al. [11] also observed a reduced yield caused by the highest dose of P (20 kg/ha P_2_O_5_) applied on leaves of pea. Contrary to the *npa* cultivars, the *lpa* line 1-2347-144 was positively affected by all fertilization variants, including P3, and provided significantly enhanced seed production compared to the variant P0. The stepped phosphorus application increased the production by 17.8% (P1), up to 22.2% (P3). The most significant effect of the foliar phosphorus application was achieved at the *lpa* line 1-150-81. In this case, the seed yield was also linearly increased by the applied phosphorus doses (*r* = 0.849; *p* ˂ 0.001), and the highest level of the P fertilization (P3) induced the enhanced yield of 42.1%. The result corresponded to the response of the low-phytate lines of soyabean on the P fertilization, where the seed yield was significantly (*p* ≤ 0.05) higher than at the normal-phytate genotypes [39]. However, these presented responses of pea on the foliar phosphorus fertilization were contradictory to the conclusions of Froese et al. [11]. In this study, the foliar P application was unable to substitute the seed-placed monoammonium phosphate and, overall, it had a marginal effect on the grain yield, P uptake as well as the seed nutritional value. According to the yield response on the phosphorus fertilization, the studied pea lines can be divided in two groups. While in the case of the *lpa* lines (1-2347-144; 1-150-81), the yield significantly correlated with the seed weight (*r* = 0.597, *p* = 0.041; *r* = 0.752, *p* = 0.005), in the case of the *npa* cultivars (CDC Bronco, and Cutlass), the pea production was significantly influenced by the number of seeds (*r* = 0.828, *p* = 0.001; *r* = 0.600, *p* = 0.039). According to the available literature, a positive influence of the foliar phosphorus application on the seed weight of common bean [39], chickpea [40], wheat, and maize [38] etc. was proved. Although the seed weight was relatively enhanced at all tested genotypes (Table 2), the influence of the phosphorus nutrition on the seed weight was significant (*p* ≤ 0.01) only at the application of the highest P dose (P3) on the *lpa* line 1-150-81. The number of seeds produced by the pea plants was also influenced by the P nutrition. After the application of the P doses P2 and P3, the number of seeds was significantly increased (*p* ≤ 0.01) at cultivars CDC Bronco and Cutlass by the mean 17.9% and 18.1%, respectively, compared to the non-fertilized variant (P0). In the case of the low-phytate lines, the number of seeds was also significantly (*p* ≤ 0.01) increased by 8.0% and 13.0%, respectively, compared to P0 (Table 2).

The foliar phosphorus application also significantly influence the height of plants, which correlated with the seed yield at the tested genotypes (*r* = 0.786, *p* ˂ 0.05). While in the case of the *lpa* pea lines, the plant height was significantly (*p* ≤ 0.01) increased even at the application of the highest phosphorus dose (P3), in the case of the cultivars CDC Bronco and Cutlass, the P3 dose did not influence the plant height. The cultivar Cutlass had no response on the P fertilization. By the mean values of the tested pea genotypes, the plant height was increased after the P fertilization by 3.4 (P1), 8.8 (P2), and 11.6 cm (P3) in the case of the low-phytate lines, and by 6.8 (P1), 7.0 (P2), and 1.4 cm (P3) in the case of the conventional cultivars (*npa*), compared to the control P0. According to the available literature, the phosphorus application to the soil had a significant influence on the height of cowpea [41]. Maize and wheat plant heights were also significantly increased by the foliar P application [38]. Interactions of the foliar P application with magnesium fertilization significantly increased the plant height of faba bean [10].

### 2.3. The Effect of Foliar P Fertilization, on Pea Seed Quality

Within the seed, P is primarily stored as phytic acid and/or phytate that accumulate in protein vacuoles. Phytate comprises up to 80% of the total seed phosphorus, and can comprise as much as 1.5% of the seed dry weight [42]. Significantly, the lowest (*p* ≤ 0.05) content of phytate in seeds was determined at the *lpa* line 1-150-81. By the P application, its content was reduced by 19.9% (P1), 24.9% (P2), and 9.1% (P3), respectively, but not significantly (*p* ≤ 0.01). Nevertheless, the line 1-150-81 was the only one that tended to the decrease of phytate in seeds by the stepped phosphorus application. The foliar P application had a limited effect on phytate in seeds of canola, wheat, and pea [11]. In that experiment, phytate concentration in the pea seeds was decreased, comparing to the P non-fertilized variant, only by the application of the P-fertilizer (10 kg/ha P_2_O_5_) that was safely placed in the seed row with pea in combination with the foliar P application (10 kg/ha P_2_O_5_). The other tested genotypes, including the *lpa* line 1-2347-144, shown a relative increase of the phytate content by the stepped P application, significantly in the cultivar Cutlass, only (Table 3). It has been reported that the phytate concentration in seeds gradually increased and was positively correlated with the applied P levels in soybean [18], oat [19], and maize [20]. In context with the phosphorus content in seeds (Table 3), which was also significantly enhanced by fertilization only in the case of the cultivar Cutlass (*p* ≤ 0.01), the portion of phytate-P amount to the total P content in seeds was evaluated for the studied genotypes (Figure 2). While in the case of the cultivar Cutlass, the fertilization by doses P2 and P3 increased the phytate-P portion from 65.6% (P0) to 73.4% and 73.7%, respectively, in the case of the line 1-150-81, a provable (*p* ≤ 0.05) decrease of the phytate-P portion from 47.8% (P0) to 40.9% (P3) was induced by the foliar P nutrition. 

The positive effect of the foliar phosphorus application on the crude protein content in the pea seeds was proved for the cultivar Cutlass, only (Table 3). In the case of the *lpa* line, content of the crude protein was not influenced by the P nutrition. The study by Klimek-Kopyra et al. [43] showed that phosphorus had a limited effect on this parameter. The phosphorus application in doses of 70 and 140 kg/ha P_2_O_5_, respectively, to the soil at 6 lines of pea did not influence content of the crude protein. Our study also does not confirm a consistent tendency towards the increase of the crude protein under higher phosphorus doses for cultivars of pea. Conversely, various *studies* [44,45] presented that the P fertilization increased the crude protein content of cowpea grain as well as low-phytate and normal-phytate cultivars of soybean [34].

The content of ash material and particular nutrients in the pea seeds of the tested genotypes is presented in Table 4. The relatively highest content of ash material was determined at the variant P3 for all tested genotypes. In this case, the mean increase of ash content was 0.13% compared to the variant P0. Besides the cultivar CDC Bronco, a significantly effect of the phosphorus application in doses P2 and/or P3 on the content of ash material was proved in tested genotypes. An important nutrient contained in the pea grain is potassium. Although its amount significantly correlated with the ash content (*r* = 0.870, *p* ˂ 0.05), the foliar phosphorus application (P3) increased its content significantly only in the case of the line 1-2347-144 (Table 4). Potassium content was increased insignificantly by application of the highest dose of phosphorus (P3), from 0.92% to 0.98% DW on average across all tested pea genotypes. By contrast, the enhancement of potassium content in the tissue of the peanut plant observed after the phosphorus application was presented by Malakondaiach and Rajeswararao [46]. A significant increase in potassium uptake due to the increasing doses of phosphorus application was also found out in cowpea [47], mung bean [48], and urd bean [49]. The content of magnesium did not change by the P fertilization in seeds of the *lpa* lines, but it was increased in the *npa* cultivars (Table 4). A diverse response of the *npa* and *lpa* genotypes on the phosphorus fertilization was observed in the case of calcium utilization, too. While its content was not influence by the P fertilization of the cultivar CDC Bronco, it was significantly enhanced in the cultivar Cutlass. In the case of the low-phytate lines, Ca content was decreased, and in the case of the line 1-150-81 significantly. Since the effect of foliar P fertilization on seed mineral content in some genotypes is only statistically significant for some treatments, it is not possible to say that foliar P application will increase seed nutritive value.

Zinc absorption capacity was reduced by the high phosphorus utilization, and zinc in the plant and soil was in a state of antagonism with phosphorus. According to the available studies, a deficiency of zinc in plants was caused by phosphorus fertilization to the soil [50,51]. The foliar phosphorus application significantly decreased the zinc content stored in seeds both in the *lpa* line (1-150-81) as well as in the *npa* cultivar (CDC Bronco). One of possible explanations can be the fact that the increased P concentration in seeds due to the foliar phosphorous application can aggravate Zn deficiency. Another explanation is that zinc concentration in seeds of pea was decreased by the effect of the induced growth response on the P fertilization. In other words, zinc was diluted in the plant tissues. Thus, the foliar phosphorus application can decrease the nutrition value of the pea seeds.

The results of the pot experiment show the possibilities of using the foliar phosphorus fertilization in the growing of normal and low-phytate pea genotypes. It is shown that the foliar application increases not only the seed yield, but also their nutritional quality, usability in the food industry and human nutrition. It is clear that the phytate content of the seeds can be regulated in this way, which is variety dependent. A decrease in phosphorus binding to phytate was obtained in the low phytate line 1-150-81 and an increase in the Canadian variety Cutlass after foliar application of phosphorus. However, verification of the results obtained from the pot experiment in field trials will be necessary.

## 3. Materials and Methods

### 3.1. Plant Materials, Plant Cultivation and Conditions of Growth

The effect of the foliar phosphorus application on photosynthetic parameters, seed yields and quality of four pea genotypes (*Pisum Sativum* L.) was investigated in this study. The experiment was conducted in growth box (PlantMaster, CLF Plant Climatics GmbH, Wertingen, Germany) at the Mendel University in Brno located at 49°21′03′′ N and 16°61′38′′ E. The low-phytate pea lines (*lpa*) 1-150-81 and 1-2347-144 [17] were chosen for this study. In these lines, the content of phytate phosphorus is reduced by approximately 60% compared to the normal phytate genotype, with a compensating increase in inorganic phosphorus [17]. Both lines were derived from the cultivar CDC Bronco [52] through chemical mutagenesis [17]. Further, the normal-phytate cultivars (*npa*) CDC Bronco and Cutlass [53] were also used for this study. Pea plants were grown under the control condition: 12 h of light (light intensity 550 μmol m^−2^ s^−1^) and 12 h of darkness; day temperature 22 °C and night temperature 15 °C; humidity 60% during the day and 90% at night. Mitscherlich pots with a volume of 6.2 L were used to grow the pea plants. Each container contained 6000 g of arable soil with a constant composition (Table 5).

Six pea seeds of each genotype per pot were sown on 27 April 2020. Four plants were grown in each pot for all pea genotypes. The foliar fertilization of phosphorus was carried out on the plant development stages of the 6th true leaf unfolded at the 6th node (3 June 2020). The following four treatments with three doses of the foliar phosphorous application were included in the experiment: P0—without P (0 mg P/pot), P1—62.5 mg P_2_O_5_ (27.3 mg P) per pot; P2—125 mg P_2_O_5_ (54.5 mg P) per pot and P3—187.5 mg P_2_O_5_ (81.8 mg P) per pot. These treatments of the foliar phosphorus application were carried out for all genotypes (for each genotype separately). The phosphorous foliar application (phosphoric acid, H₃PO₄) in 5 mL of water solution per pot at each treatment (P1−P3) was used. The control variant (P0) was treated with water. Water and phosphorus solutions were regularly applied using a pressurized hand pump sprayer (DPZ 1500, ProGlass, Weilheim an der Teck, Germany). The vegetation pots were arranged randomly in the growth box. A total of 32 pots were established for each genotype: four treatments with the graded dose of P (P0–P3); each treatment was established in eight replications (pots).

### 3.2. Measurement of Photosynthetic Parameters of Pea Plants

Selected photosynthetic parameters of pea plant The content of chlorophyll (like N-tester value), and selected parameters of fluorescence (quantum yield of photosystem II, and chlorophyll fluorescence decrease ratio) were evaluated. The measurements were performed 14 (T1) and 28 days (T2) after the foliar application.

#### 3.2.1. Content of Chlorophyll (N-Tester Value)

The content of chlorophyll in pea leaves, expressed as N-tester values, was determined using an N-Tester instrument (Yara International ASA, Oslo, Norway) in the wavelength range 650–940 nm [55]. Chlorophyll content was determined from leaves located in the middle part of the plants.

#### 3.2.2. Chlorophyll Fluorescence Parameters

Photochemical *efficiency* of *photosystem II* (PSII) in pea plants was determined. Chlorophyll fluorescence determination was performed with a PAR-FluorPen FP 110-LM/S (Photon Systems Instruments, Drásov, Czech Republic). The measured data were subsequently evaluated using FluorPen 1.1 software [56]. The leaves of the pea plant were dark adapted for 30 min prior to the measurement. Protocol for the measurement of chlorophyll fluorescence parameters is presented in Table 6.

The Quantum yield of the PSII (*Φ_PSII_*) and the Chlorophyll fluorescence decrease ratio (*R_Fd_*) were measured as a photochemical quenching parameter (Table 7).

### 3.3. Yield Parameters and Seed Quality

The pea plants were harvested at the full ripeness on 29 July 2020. The height of plants, the yield of seeds per pot, the weight of one thousand seeds, and the number of seeds were determined. Plant height was measured before harvesting. After cutting the plants, the pea seeds were harvested by hand. The pea seeds were then weighed (laboratory scale PCB Kern, KERN and Sohn GmbH, Balingen, Germany), and counted (seed counter Contador, Pfeuffer GmbH, Kitzingen, Germany). The weight of 1000-seeds was subsequently determined.

Subsequently, the pea seeds were analyzed in order to evaluate selected qualitative parameters, namely: the content of phytic acid (phytate), content of phosphorus in the phytate, content of crude protein, ash material, and content of nutrients in seeds (P, K, Mg, Ca, and Zn).

#### 3.3.1. Determination of Phytic Acid (Phytate)

Content of phytic acid was determined using a commercial kit „Phytic acid (phytate)/Total phosphorus“ (Megazyme, Bray, Ireland) [59]. Pea seeds were finely ground using the Foss Tecator Cyclotec 1093 (Foss Analytical, Hillerød, Denmark). For analysis, 1 g of flour was weighted. Phytic acid from the sample was extracted with 0.66 M HCl. The neutralized aliquot of the sample was treated with phytase that was specific for phytic acid, and the lower myo-inositol phosphate forms. Then, the sample was treated with alkaline phosphatase that hydrolyzed myo-inositol phosphates and released free phosphate. The total phosphate was measured using a colorimetric method with molybdenum blue (Spekol 1300, Analytik Jena AG, Jena, Germany). The amount of molybdenum blue was proportional to the amount of phosphate in the sample.

#### 3.3.2. Analysis of Crude Protein and Ash

Ash material was determined by weighing the material remaining after the burning of a fixed-weight sample at 550 °C and specified conditions. Nitrogenous substances were measured by the Kjeldal method (N × 6.25 coefficient) using the Kjeltec 2300 device (Foss Analytical, Hillerød, Denmark) [60].

#### 3.3.3. Determination of Nutrient Contents

The samples of pea seeds were dried at temperature of 50 °C, then crushed in the grinder (Foss Tecator Cyclotec 1093, Foss Analytical, Hillerød, Denmark), and homogenized. After the microwave closed vessel acid digestion (HNO_3_/H_2_O_2_) in ETHOS One (Milestone Srl, Sorisole, Italy), the contents of nutrients were determined (Table 8).

### 3.4. Statistical Data Analysis

Measured data were statistically evaluated using STATISTICA 12 program (TIBCO Software Inc., Palo Alto, CA, USA) [62]. The normality and homogeneity of variances were verified, respectively, by Shapiro-Wilk’s and Levene’s test at *p* ≤ 0.05. The influence of the monitored factors was analyzed via two-way ANOVA (level of significance *p* ≤ 0.05). All evaluated parameters are expressed in tables and graphs as the arithmetic mean ± standard deviation (SD). The differences between the arithmetical means were evaluated by the Fisher’s (*LSD*) test at the 95% (*p* < 0.05), and 99% (*p* < 0.01) level of significance.

## Figures and Tables

**Figure 1 plants-10-01608-f001:**
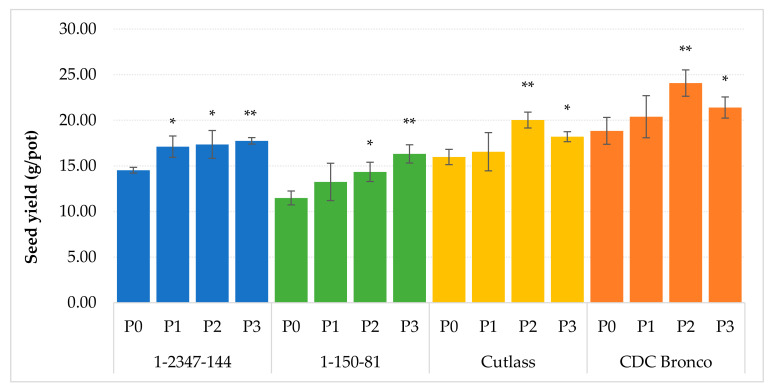
The effect of the foliar phosphorus application on the seed yield. The mean values (*n* = 8) marked with an asterisk are significantly different (* *p* ≤ 0.05; ** *p* ≤ 0.01) from the variant without the P fertilization (P0) by the Fisher’s LSD test (each of the genotypes was statistically evaluated separately). Error bars represent the standard deviation of arithmetical mean (SD).

**Figure 2 plants-10-01608-f002:**
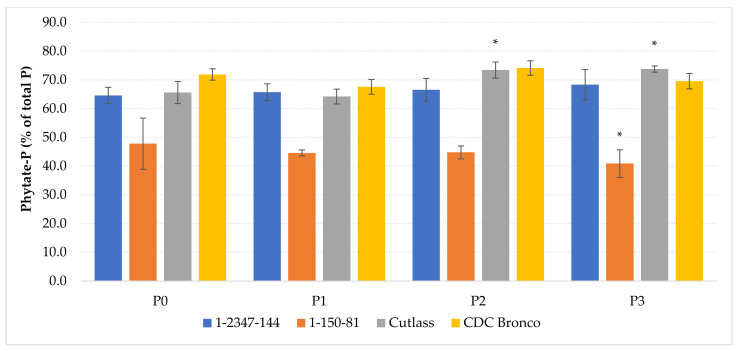
The effect of foliar phosphorus application on portion of phytate-P amount to the total P content in seeds. The mean values (*n* = 8) marked with an asterisk are significantly different (* *p* ≤ 0.05) from the variant without P fertilization (P0) by the Fisher’s LSD test (each of the genotypes was statistically evaluated separately). Error bars represent the standard deviation of arithmetical mean (SD).

**Table 1 plants-10-01608-t001:** The effect of the foliar phosphorus application on chlorophyll contents (N-tester value) and chlorophyll fluorescence parameters (*Φ_PSII_* and *R_Fd_*).

Genotype	Treatment	N-Tester Value		*Φ_PSII_*		*R_Fd_*	
		T1	T2	T1	T2	T1	T2
1-2347-144	P0	448 ± 6	378 ± 5	0.801 ± 0.011	0.809 ± 0.014	2.04 ± 0.03	1.94 ± 0.04
	P1	458 ± 5	390 ± 4 *	0.820 ± 0.008 *	0.825 ± 0.010	2.13 ± 0.02	2.06 ± 0.05 *
	P2	464 ± 7 **	401 ± 4 **	0.818 ± 0.007 *	0.831 ± 0.004	2.42 ± 0.20 **	2.07 ± 0.04 *
	P3	463 ± 6 *	402 ± 3 **	0.822 ± 0.002 *	0.833 ± 0.007 *	2.42 ± 0.20 **	2.13 ± 0.07 **
1-150-81	P0	443 ± 10	371 ± 3	0.792 ± 0.015	0.803 ± 0.004	1.99 ± 0.02	1.93 ± 0.07
	P1	456 ± 10 *	383 ± 7 *	0.816 ± 0.004 **	0.813 ± 0.005	2.08 ± 0.04	2.00 ± 0.07
	P2	464 ± 7 **	373 ± 9	0.816 ± 0.008 **	0.837 ± 0.005 **	2.11 ± 0.02	2.08 ± 0.06 *
	P3	475 ± 13 **	401 ± 7 **	0.829 ± 0.005 **	0.840 ± 0.007 **	2.16 ± 0.04 *	2.10 ± 0.04 **
Cutlass	P0	439 ± 6	368 ± 8	0.803 ± 0.011	0.792 ± 0.018	2.06 ± 0.04	1.77 ± 0.09
	P1	452 ± 5 *	374 ± 6	0.823 ± 0.008 *	0.792 ± 0.009	2.02 ± 0.03	1.82 ± 0.07
	P2	458 ± 5 **	402 ± 3 **	0.829 ± 0.005 **	0.815 ± 0.008	2.09 ± 0.06	1.97 ± 0.06 **
	P3	458 ± 6 **	381 ± 5 *	0.818 ± 0.019	0.789 ± 0.003	2.09 ± 0.02	1.76 ± 0.05
CDC Bronco	P0	447 ± 5	386 ± 8	0.804 ± 0.015	0.812 ± 0.043	2.00 ± 0.07	1.91 ± 0.04
	P1	463 ± 5 *	394 ± 12	0.828 ± 0.004 **	0.827 ± 0.006	2.05 ± 0.10	1.94 ± 0.07
	P2	465 ± 8 **	408 ± 4 **	0.825 ± 0.007 *	0.816 ± 0.004	2.22 ± 0.08 **	2.02 ± 0.08
	P3	468 ± 6 **	398 ± 6 *	0.820 ± 0.011 *	0.777 ± 0.008 **	2.07 ± 0.07	1.86 ± 0.13

The mean values marked with asterisk are significantly different (* *p* ≤ 0.05; ** *p* ≤ 0.01) from the variant without P fertilization (P0) by Fisher’s LSD test (each of the genotypes was statistically evaluated separately). The values in the table represent the arithmetic mean (*n* = 60) ± SD (standard deviation). Determinations were carried out 14 (T1) and 28 (T2) days after the P-application.

**Table 2 plants-10-01608-t002:** The effect of the foliar phosphorus application on the plant height, seed weight and number of seeds.

Genotype	Treatment	Plant Height (cm)	Seed Weight (g/1000 Seeds)	Seed Number (Seed/Pot)
1-2347-144	P0	63 ± 5	193 ± 10	75.3 ± 3.8
	P1	66 ± 9	209 ± 20	81.7 ± 0.6 **
	P2	72 ± 5 *	220 ± 29	79.0 ± 2.0
	P3	75 ± 8 **	212 ± 16	84.0 ± 1.0 **
1-150-81	P0	64 ± 7	185 ± 10	62.3 ± 1.5
	P1	69 ± 7	197 ± 30	67.3 ± 2.5 *
	P2	73 ± 17 *	207 ± 26	69.3 ± 3.1 **
	P3	76 ± 9 **	229 ± 22 **	71.3 ± 1.5 **
Cutlass	P0	72 ± 2	188 ± 20	85.0 ± 4.4
	P1	75 ± 7	217 ± 7	76.3 ± 1.5 **
	P2	73 ± 4	212 ± 16	94.3 ± 4.5 **
	P3	71 ± 3	181 ± 16	100.7 ± 2.3 **
CDC Bronco	P0	98 ± 18	234 ± 16	80.3 ± 1.5
	P1	109 ± 11 **	238 ± 9	86.0 ± 3.6 *
	P2	111 ± 9 **	240 ± 10	100.3 ± 2.5 **
	P3	102 ± 9	226 ± 25	94.7 ± 1.5 **

The mean values marked with an asterisk are significantly different (* *p* ≤ 0.05; ** *p* ≤ 0.01) from the variant without P fertilization (P0) by the Fisher’s LSD test (each of the genotypes was statistically evaluated separately). The values in the table represent the arithmetic mean (*n* = 8) ± SD (standard deviation).

**Table 3 plants-10-01608-t003:** The effect of the foliar phosphorus application on the content of phosphorous, phytate, and crude protein in pea seeds.

Genotype	Treatment	P (% DM)	Phytate (g/100 g DM)	Crude Protein (% DM)
1-2347-144	P0	0.36 ± 0.02	0.82 ± 0.07	22.2 ± 0.4
	P1	0.35 ± 0.02	0.82 ± 0.04	23.4 ± 0.7
	P2	0.38 ± 0.06	0.90 ± 0.19	22.6 ± 0.4
	P3	0.41 ± 0.08	1.00 ± 0.27	23.3 ± 2.0
1-150-81	P0	0.42 ± 0.02	0.82 ± 0.22	23.7 ± 0.8
	P1	0.42 ± 0.04	0.66 ± 0.07	22.4 ± 2.0
	P2	0.39 ± 0.01	0.62 ± 0.04	20.9 ± 1.6 **
	P3	0.45 ± 0.02	0.75 ± 0.19	21.9 ± 0.4
Cutlass	P0	0.30 ± 0.01	0.69 ± 0.05	20.1 ± 0.5
	P1	0.30 ± 0.02	0.69 ± 0.06	21.1 ± 0.3
	P2	0.39 ± 0.03 **	1.01 ± 0.10 **	22.7 ± 1.1 **
	P3	0.43 ± 0.03 **	1.13 ± 0.08 **	24.1 ± 1.2 **
CDC Bronco	P0	0.41 ± 0.03	1.04 ± 0.10	20.0 ± 0.2
	P1	0.41 ± 0.03	0.98 ± 0.07	20.4 ± 1.1
	P2	0.41 ± 0.04	1.08 ± 0.07	20.7 ± 0.9
	P3	0.46 ± 0.03	1.14 ± 0.10	20.7 ± 0.8

The mean values marked with an asterisk are significantly different (** *p* ≤ 0.01) from the variant without P fertilization (P0) by the Fisher’s LSD test (each of the genotypes was statistically evaluated separately). The values in the table represent the arithmetic mean (*n* = 8) ± SD (standard deviation). (DM) dry matter.

**Table 4 plants-10-01608-t004:** The effect of the foliar phosphorus application on ash content, and mineral content in pea seeds.

Genotype	Treatment	Ash (% DM)	K (% DM)	Mg (% DM)	Ca (% DM)	Zn (mg/kg DM)
1-2347-144	P0	2.89 ± 0.08	0.92 ± 0.02	0.125 ± 0.005	0.036 ± 0.005	34.0 ± 5.3
	P1	2.84 ± 0.06	0.85 ± 0.07	0.118 ± 0.011	0.041 ± 0.003	29.8 ± 3.2
	P2	2.96 ± 0.17	0.92 ± 0.05	0.121 ± 0.005	0.033 ± 0.003	25.8 ± 2.3
	P3	3.14 ± 0.22 *	1.01 ± 0.09 *	0.125 ± 0.009	0.030 ± 0.004	26.0 ± 2.3
1-150-81	P0	3.26 ± 0.13	1.02 ± 0.04	0.121 ± 0.008	0.040 ± 0.007	28.7 ± 0.6
	P1	3.13 ± 0.29	0.99 ± 0.09	0.114 ± 0.010	0.035 ± 0.003	28.5 ± 1.2
	P2	2.97 ± 0.06 *	0.95 ± 0.00	0.112 ± 0.006	0.032 ± 0.007 *	26.6 ± 5.8 *
	P3	3.09 ± 0.04	1.01 ± 0.00	0.121 ± 0.004	0.030 ± 0.004 **	25.6 ± 1.2 *
Cutlass	P0	2.68 ± 0.06	0.82 ± 0.06	0.109 ± 0.002	0.032 ± 0.002	24.7 ± 2.4
	P1	2.68 ± 0.07	0.83 ± 0.01	0.114 ± 0.009	0.033 ± 0.005	25.6 ± 4.0
	P2	2.96 ± 0.06 *	0.90 ± 0.03	0.116 ± 0.002	0.035 ± 0.005	25.6 ± 3.7
	P3	3.02 ± 0.18 **	0.87 ± 0.08	0.129 ± 0.007 **	0.045 ± 0.006 **	25.2 ± 1.6
CDC Bronco	P0	2.96 ± 0.14	0.93 ± 0.07	0.104 ± 0.004	0.030 ± 0.002	34.9 ± 3.7
	P1	2.96 ± 0.08	0.96 ± 0.03	0.111 ± 0.004	0.029 ± 0.002	34.4 ± 4.1
	P2	3.02 ± 0.11	0.96 ± 0.04	0.116 ± 0.007 *	0.029 ± 0.002	27.1 ± 2.4 *
	P3	3.08 ± 0.01	1.01 ± 0.01	0.108 ± 0.003	0.031 ± 0.005	29.1 ± 4.2

The mean values marked with an asterisk are significantly different (* *p* ≤ 0.05; ** *p* ≤ 0.01) from the variant without P fertilization (P0) by the Fisher’s LSD test (each of the genotypes was statistically evaluated separately). The values in the table represent the arithmetic mean (*n* = 8) ± SD (standard deviation).

**Table 5 plants-10-01608-t005:** Chemical composition of soil used in this study.

Soil Parameter	Value
pH (CaCl_2_)	6.09
Soil oxidizable carbon (Cox)	0.80%
Clay	20%
Silt	27%
Sand	53%
CEC (Cation Exchange Capacity)	164 mmol/kg
total N	0.19%
Ammonium N (NH_4_^+^)	1.48 mg/kg
Nitrate N (NO_3_^−^)	17.2 mg/kg
Available P (Mehlich III)	36.4 mg/kg *
Available K (Mehlich III)	400 mg/kg
Available Ca (Mehlich III)	2 720 mg/kg
Available Mg (Mehlich III)	214 mg/kg

* low available phosphorus content. Soil parameters were determined according to Zbíral [54].

**Table 6 plants-10-01608-t006:** Measurement protocol of the chlorophyll fluorescence.

Chlorophyll Fluorescence Parameters	Pulse Type	Light Intensity (μmol/m^2^/s)	Phase	Duration (s)	1st Pulse (s)	Pulse Interval (s)
*Φ* *_PSII_*	Saturation	2400	-	1 pulse		
*R_Fd_*	Flash	900	L	60	0.2	1
			DR	88	1	1
	Saturation	2400	L	60	7	12
			DR	88	11	26
	Actinic	300	L	60	-	-

*λ* = 454 nm, L—light, DR—Dark recovery.

**Table 7 plants-10-01608-t007:** The photochemical quenching parameter.

Chlorophyll Fluorescence Parameters		Ref.
*Φ* *_PSII_*	F_m_ − F_0_/F_m_	[57]
*R_Fd_*	F_d_/F_s_	[58]

(F_0_) minimal fluorescence from the dark-adapted leaves, F_m_ means maximal fluorescence from the dark-adapted leaves; (F_d_) fluorescence decrease from (F_m_ to F_s_; F_s_) steady state chlorophyll fluorescence.

**Table 8 plants-10-01608-t008:** The methods for the determination of nutrients in pea seed.

Nutrient	Method Used	Device Used	Ref.
P	Spectrophotometry	Unicam 8625 UV/VIS (Pye Unicam Ltd., Cambridge, UK)	[61]
K, Mg, Ca, Zn	Atomic absorption spectrometry	ContrAA 700 (Analytik Jena AG, Jena, Germany)	[61]

(UV/VIS) ultraviolet–visible.

## Data Availability

The data presented in this study are available on request from the corresponding author. Due to the nature of this research, participants of this study did not agree for their data to be shared publicly, so supporting data is not available.

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
