# Peer review of "Response of Normal and Low-Phytate Genotypes of Pea (Pisum sativum L.) on Phosphorus Foliar Fertilization"

_plants, 2021, doi:10.3390/plants10081608_

Round 1

Reviewer 1 Report

Since, the pea has a strongly developed root, with numerous lateral branches on which the nodules are found, and the area of the root brushes has a high solubilization capacity, which allows the plant to use phosphorus and other nutrients from sparingly soluble compounds, it should be specified if the positive differences of the determined indicators are due only to the foliar fertilization or also due to the phosphorus absorbed from the soil.

Author Response

Dear reviewer,

thank you for your very careful review of our paper, and for the comments, corrections, and suggestions. We have tried to take most of them into account when revising the manuscript. We believe that this has significantly improved our manuscript. Specific responses to the comments are provided below; the revised manuscript is attached.

Reviewer's comment:

Since, the pea has a strongly developed root, with numerous lateral branches on which the nodules are found, and the area of the root brushes has a high solubilization capacity, which allows the plant to use phosphorus and other nutrients from sparingly soluble compounds, it should be specified if the positive differences of the determined indicators are due only to the foliar fertilization or also due to the phosphorus absorbed from the soil.

Authors' response:

To investigate the effect of soil phosphorus supply on growth, yield and seed quality of pea, the P0 variant (also a control variant for foliar application of P) was established. On this variant, phosphorus was supplied to the plants only from the soil supply. It should be added that the soils were homogenized at the establishment of the experiment and used in the same volume and composition (Table 5) for all treatments.

Reviewer 2 Report

The paper by Skarpa et al investigates the effect of P foliar fertilization on both normal- and low-phytate genotypes of pea. The study was performed at four levels of P. Taking into consideration that phytic acid is the main reserve of P in legumes, and that phytates had nutritional repercussions, the subject is interesting for the general reader, and it is particularly within the scope of the especial issue “Improving Fertilizer Use Efficiency-Methods and Strategies for the Future”.

The main weakness of this study is the lack of an in-deep discussion of the results. In its present form, the paper is mainly descriptive. The combination of Results and Discussion in one section has not been a good idea; the authors comment particular details of the results, but not the global significance of the study. Many questions need to be answered before the article is ready for publication.

ABSTRACT

-The objectives of the study should be clearly defined. Why were normal and low phytate genotypes used? Was it expected a rise or a decrease in phytate content of the seeds due to foliar P fertilization? Do the results of the paper show improved efficiency of P foliar fertilization with respect to soil fertilization?

-The abstract should finish with a sentence highlighting the significance of the results of the study.

-Do not use “provably” for “statistically significant” (or simply “significant”)

INTRODUCTION

-The introduction describes both beneficial and undesired effects of phytates on human nutrition and plant physiology. Has the study been designed to find a tool for decreasing phytate levels in pea seeds? Or, otherwise, does the study explore the efficiency of foliar fertilization with respect to soil fertilization? Which levels of phytates are desirable in pea seeds?

- Why were normal and low phytate genotypes used?

-At the end of the second paragraph, the author say: “A positive correlation between the level of the phosphorous fertilization and the decrease of phytate concentration was observed….” Does it mean a NEGATIVE correlation between both parameters? In addition in RESULTS, 2.3. The effect of foliar P fertilization on pea seed quality, the authors say: “It has been reported that the phytate concentration in seeds gradually increased with the applied P levels…”. This discrepancy needs to be corrected or explained.

-In the last paragraph, the authors say: “A positive effect of the foliar P application on pea seed production and its quality, especially the portion of P bounded to phytate, is expected”. A raise or a decrease? Should the result be different in normal and in low phytate phenotypes? Which are the most probably repercussions on seed physiology?

RESULTS AND DISCUSSION

2.1. Chlorophyll and fluorescence

-Why were chlorophylls quantified? Chlorosis is not a usual symptom of P deficiency.

-Why were fluorescence parameters measured instead of a direct measurement of net photosynthesis?

-Do not use “provably” for “statistically significant” (or simply “significant”)

2.2. Yield parameters

-The authors discuss the potential of foliar fertilization in comparison to soil fertilization. If this is the main goal of the study, soil fertilization should have been included in the experimental design.

-At the end of the first (and too long) paragraph, the authors say: “While in the case of the lpa lines….the yield significantly correlated with the seed weight….., in the case of the npa cultivars…..the pea production was provably influenced by the numbers of seeds”. Nevertheless, the results show that 1) P foliar fertilization increased seed yield in most conditions (Figure 1); 2) seed weight is not changed in most of conditions (Table 2); 3) Seed number is increased in most of conditions (Table 2). Why have the authors include the data of correlation, which are in conflict with the results of Figure 1 and Table 2?

-The statistical analysis of differences between genotypes is not shown. For example, the comparison of seed yield in control or fertilized conditions. Seed yield seems to be higher in normal than in low phytate genotypes. The same with treatments; for example, seed number is higher in normal than in low-phytate genotypes with P3 level of fertilization. In the same lane, there is not a statistical analysis of differences within treatments.

2.3. Seed quality

-The accurate analysis of seed quality needs validated reference values. Which are optimum levels of phytates in pea seeds?

-The results of Table 3 are discussed without referring to Table 3 in the text.

-The amount of phytates shown in Table 3 do not allow classifying the genotypes in “Normal” and “low” phytate.

-The effect of P foliar fertilization on phytate, protein, ash and mineral content of seeds is satatstically significant exclusively for certain treatments for some of the genotypes. Moreover, in the few cases when the differences were significant, it is difficult to evaluate the physiological relevance of the effect. For example, with respect to K content, the only significant variation was for 1-2347-144 with P3, and the difference was from 0.92 to 1.01 %.

-In my opinion, the results show that P foliar fertilization increases seed yield of pea, with minor changes of seed quality.

CONCLUSIONS

-Conclusions are a summary of results, without highlighting their significance.

-Some of the conclusions are not reliable. The authors say that: “Responses of the low-phytate lines on the fertilization were more remarkable due to a significant increase of the seed production even at the highest P dose which reduced the seed production of the normal-phytate genotype”. This is only true when the effect of P2 is compared with P3·, but not when P3 is compared with P0 (Figure 1). With respect to control, P3 increased seed yield and seed number in all the genotypes. In addition, at P3 the seed yield and number was higher in normal-phytate genotypes.

Author Response

Dear reviewer,

thank you for your very careful review of our paper, and for the comments, corrections, and suggestions. We have tried to take most of them into account when revising the manuscript. We believe that this has significantly improved our manuscript. Specific responses to the comments are provided below; the revised manuscript is attached.

Reviewer's comment:

The main weakness of this study is the lack of an in-deep discussion of the results. In its present form, the paper is mainly descriptive. The combination of Results and Discussion in one section has not been a good idea; the authors comment particular details of the results, but not the global significance of the study. Many questions need to be answered before the article is ready for publication.

Authors' response:

The combination of results and discussion is deliberate, mainly because of the limited literature evaluating the effect of foliar nutrition in peas. In particular, the response of low phytate genotypes to fertilization is very poorly described in the literature. It is thus very difficult to discuss the results obtained in depth.

The pot experiment was set up to study the effect of phytate content in pea seeds, for example, by foliar application of phosphorus, which follows from the basic metabolic nature of seed storage formation for the germination stage. For a more extensive discussion we need results from the field experiments, which will also be included in future publications.

ABSTRACT

Reviewer's comment:

The objectives of the study should be clearly defined. Why were normal and low phytate genotypes used? Was it expected a rise or a decrease in phytate content of the seeds due to foliar P fertilization? Do the results of the paper show improved efficiency of P foliar fertilization with respect to soil fertilization?

Authors' response:

The hypothesis is that foliar application of P will increase pea yield and seed quality in plants grown under P deficient conditions. Normal and low phytate genotypes were deliberately included in the experiment to determine their response to fertilization.

The experiment does not compare the possible ways of phosphorus fertilization: soil application and foliar application. It offers an alternative approach to optimise phosphorus nutrition of peas, i.e. the use of foliar fertilisation with this nutrient, which is a suitable procedure especially in conditions where soil application seems ineffective (e.g. inappropriate soil pH, which immobilises phosphorus in the soil).

The aim of the study is to compare mutant low phytate lines (lpa) and conventional pea varieties (npa) in terms of their ability to respond to an alternative application method (foliar application) of nutrient, in this case phosphorus. An increase of free P (inorganic) for nutritional purposes is expected. At the same time, it is being determined whether the low-phytate lines will use the P supplied by fertilization directly for binding to phytate, i.e. will this increase the P-phytate content or will only increase the total P content of the seeds and the phytate level will remain unchanged.

Reviewer's comment:

The abstract should finish with a sentence highlighting the significance of the results of the study.

Authors' response:

The abstract was completed as recommended.

Reviewer's comment:

Do not use “provably” for “statistically significant” (or simply “significant”)

Authors' response:

The manuscript was revised on the basis of the recommendations.

INTRODUCTION

Reviewer's comment:

The introduction describes both beneficial and undesired effects of phytates on human nutrition and plant physiology. Has the study been designed to find a tool for decreasing phytate levels in pea seeds? Or, otherwise, does the study explore the efficiency of foliar fertilization with respect to soil fertilization? Which levels of phytates are desirable in pea seeds?

Authors' response:

The study is not aimed at comparing the effectiveness of foliar P application with its soil application. The experiment evaluates exclusively foliar application and its effect on production and quality of seed of selected pea genotypes. Treatments with soil application of this nutrient have not been established. However, this method of fertilization is widespread and cannot be neglected, which is why it is mentioned in the manuscript.

Phytic acid is the major storage form of phosphorous in cereals, legumes, oil seeds and nuts. Phytic acid is known as a food inhibitor which chelates micronutrient (especially iron and zinc) and prevents it to be bioavailable for monogastric animals, including humans, because they lack enzyme phytase in their digestive tract. Several methods have been developed to reduce the phytic acid content in food and improve the nutritional value of cereal which becomes poor due to such antinutrient. These include genetic improvement as well as several pre-treatment methods such as fermentation, soaking, germination and enzymatic treatment of grains with phytase enzyme. Biofortification of staple crops using foliar application can potentially help in alleviating malnutrition in developing countries.

Reviewer's comment:

Why were normal and low phytate genotypes used?

Authors' response:

As already mentioned, biofortification (foliar application) is one of the methods to increase not only the yield, but especially the quality of plant products (food, feed). However, there is little information in the available literature on the response of peas to foliar phosphorus fertilisation. One possibility is the use of low phytate pea lines where foliar application of P can also be used. Our aim was to determine the response of not only normal but also low phytate pea genotypes to foliar application of P.

We had the opportunity to test lines with a genetically fixed low phytate phenotype. However, we do not know the response of these genotypes to foliar phosphorus fertilization in terms of changes in the content of the observed P forms.

Reviewer's comment:

At the end of the second paragraph, the author say: “A positive correlation between the level of the phosphorous fertilization and the decrease of phytate concentration was observed….” Does it mean a NEGATIVE correlation between both parameters? In addition in RESULTS, 2.3. The effect of foliar P fertilization on pea seed quality, the authors say: “It has been reported that the phytate concentration in seeds gradually increased with the applied P levels…”. This discrepancy needs to be corrected or explained.

Authors' response:

The manuscript was edited according to the recommendations.

Reviewer's comment:

In the last paragraph, the authors say: “A positive effect of the foliar P application on pea seed production and its quality, especially the portion of P bounded to phytate, is expected”. A raise or a decrease?

Authors' response:

The manuscript was edited according to the recommendations.

Reviewer's comment:

Should the result be different in normal and in low phytate phenotypes?

Authors' response:

In principle, the low-phytate phenotype may not have a higher total P. Phosphorus fertilisation (biofortification of P) is expected to change the ratio between inorganic (unbound) phosphorus and phosphorus bound in phytate. The differences between mutant pea lines (lpa) and their parent varieties (npa) are shown in the literature cited in the section “Materials and Methods”.

Warkentin, T.D.; Delgerjav, T.; Arganosa, G.; Rehman, A.U.; Bett, K.E.; Anbessa, Y.K; Rossnagel, B.; Raboy, V. Development and characterization of low-phytate pea. Crop Sci. 2012, 52, 74–78.

Warkentin, T.; Vandenberg, A.; Banniza, S.; Slinkard, A. CDC Bronco field pea. Can. J. Plant Sci. 2005, 85, 649–650.

Warkentin, T.D.; Delgerjav, O.; Arganosa, G.; Rehman, A.U.; Bett, K.E.; Anbessa, Y.; Rossnagel, B.; Raboy, V. Development and Characterization of Low-Phytate Pea. Crop. Sci. 2012, 52, 74–78.

Blade, S.; Warkentin T.; Vandenberg, A. Cutlass field pea. Can. J. Plant Sci. 2004, 84(2), 533-534.

The available literature indicates that the low-phytate lines had grain yields 8-14% lower than that of normal phytate varieties on average, and their seed weight was 6% lower. This is probably also related to the accumulation of phytate in the seeds, in other words, phytate accumulation is probably yield dependent.

RESULTS AND DISCUSSION

Reviewer's comment:

Why were chlorophylls quantified? Chlorosis is not a usual symptom of P deficiency.

Authors' response:

Although chlorosis is not a typical symptom of phosphorus deficiency, phosphorus has a significant effect on photosynthesis, as shown in a study CARSTENSEN, Andreas, Andrei HERDEAN, Sidsel Birkelund SCHMIDT, Anurag SHARMA, Cornelia SPETEA, Mathias PRIBIL a Søren HUSTED. The Impacts of Phosphorus Deficiency on the Photosynthetic Electron Transport Chain. Plant Physiology [online]. 2018, 177(1), 271-284 [cit. 2021-7-18].  (doi:10.1104/pp.17.01624).

The direct influence of the phosphorus content of the plant on the chlorophyll content is confirmed by studies such as.

  1. Valentine, B. Osborne, D. Mitchell Interactions between phosphorus supply and total nutrient availability on mycorrhizal colonization, growth and photosynthesis of cucumber Scientia Horticulturae, 88 (2001), pp. 177-189;

Siedliska, A., Baranowski, P., Pastuszka-Woźniak, J. et al. Identification of plant leaf phosphorus content at different growth stages based on hyperspectral reflectance. BMC Plant Biol 21, 28 (2021). https://doi.org/10.1186/s12870-020-02807-4;

Meng X, Chen W-W, Wang Y-Y, Huang Z-R, Ye X, Chen L-S, et al. (2021) Effects of phosphorus deficiency on the absorption of mineral nutrients, photosynthetic system performance and antioxidant metabolism in Citrus grandis. PLoS ONE 16(2): e0246944. https://doi.org/10.1371/journal.pone.0246944

Reviewer's comment:

Why were fluorescence parameters measured instead of a direct measurement of net photosynthesis?

Authors' response:

The evaluation of the effect of foliar application of phosphorus on plants was carried out using methods available at our department.  These include instruments for measuring fluorescence parameters. These parameters measure the pro-portion of the light absorbed by chlorophyll associated with the PSII. Measurement of chlorophyll fluorescence parameters is a frequently used method for detecting plant stress caused by biotic and abiotic influences. Measurement of net photosynthesis was not successful in this experiment.

Do not use “provably” for “statistically significant” (or simply “significant”)

Authors' response:

The manuscript was edited according to the recommendations.

Reviewer's comment:

The authors discuss the potential of foliar fertilization in comparison to soil fertilization. If this is the main goal of the study, soil fertilization should have been included in the experimental design.

Authors' response:

As already mentioned, the soil phosphorus fertilization was not included in the experimental design. However, soil application of phosphorus is an essential means of optimizing plant nutrition with this nutrient and was therefore discussed in the manuscript.

Reviewer's comment:

At the end of the first (and too long) paragraph, the authors say: “While in the case of the lpa lines….the yield significantly correlated with the seed weight….., in the case of the npa cultivars…..the pea production was provably influenced by the numbers of seeds”. Nevertheless, the results show that 1) P foliar fertilization increased seed yield in most conditions (Figure 1); 2) seed weight is not changed in most of conditions (Table 2); 3) Seed number is increased in most of conditions (Table 2). Why have the authors include the data of correlation, which are in conflict with the results of Figure 1 and Table 2?

Authors' response:

The graph and table show the average values of the observed characteristics. Correlation analysis was used to determine the relationship between seed yield (g/pot) and yield-forming factors (yield components) (seed weight - g/1000 seeds and seed number - seed/pot). The correlation was found from all repetitions (each fertilization variant was repeated 8 times for each genotype). The correlation for each variety is given below.

It shows that a significant dependence of yield on seed weight was found in the case of lpa, in the case of npa it was not conclusive (it was low).

1-2347-144       yield/number of seeds 0.7120 (p=0.009); yield/seed weight 0.5968 (p=0.041)

1-150-81          yield/seed number 0.6461 (p=0.023); yield/seed weight 0.7524 (p=0.005)

CDC BRONCO yield/seed number 0.8278 (p=0.001); yield/seed weight 0.1351 (p=0.676)

CULTASS      yield/seed number 0.6001 (p=0.039); yield/seed weight 0.1141 (p=0.724)

Reviewer's comment:

The statistical analysis of differences between genotypes is not shown. For example, the comparison of seed yield in control or fertilized conditions. Seed yield seems to be higher in normal than in low phytate genotypes. The same with treatments; for example, seed number is higher in normal than in low-phytate genotypes with P3 level of fertilization. In the same lane, there is not a statistical analysis of differences within treatments.

Authors' response:

The aim of the study was not to compare individual genotypes (or groups of genotypes) with each other. Our interest was to verify the effect of phosphorus fertilization on seed yield and its quality. Research results that focus on comparing low phytate lines are published.

WARKENTIN, T. D., O. DELGERJAV, G. ARGANOSA, A. U. REHMAN, K. E. BETT, Y. ANBESSA, B. ROSSNAGEL a V. RABOY. Development and Characterization of Low-Phytate Pea. Crop Science [online]. 2012, 52(1), 74-78 [cit. 2021-7-20]. doi:10.2135/cropsci2011.05.0285

SHUNMUGAM, Arun, Cheryl BOCK, Gene ARGANOSA, Fawzy GEORGES, Gordon GRAY a Thomas WARKENTIN. Accumulation of Phosphorus-Containing Compounds in Developing Seeds of Low-Phytate Pea (Pisum sativum L.) Mutants. Plants [online]. 2015, 4(1), 1-26 [cit. 2021-7-20].10.3390/plants4010001

WARKENTIN, Tom, Nikolai KOLBA a Elad TAKO. Low Phytate Peas (Pisum sativum L.) Improve Iron Status, Gut Microbiome, and Brush Border Membrane Functionality In Vivo (Gallus gallus). Nutrients [online]. 2020, 12(9) [cit. 2021-7-20]. doi:10.3390/nu12092563

Reviewer's comment:

The accurate analysis of seed quality needs validated reference values. Which are optimum levels of phytates in pea seeds?

Authors' response:

The primary functions of phytates in seeds are storage of phosphates as energy source and antioxidant for the germinating seed. Phytate is also an important mineral reserve in seeds, and it is stored in protein storage vacuoles in the aleurone cell layer or the embryo of the seed. Phytates are also involved in stress responses, membrane biogenesis and intracellular signalling. For this purpose, it is advisable to grow normal or high phytate pea genotypes.

On the other hand, from the point of view of animal (and therefore human) nutrition, it is exactly the opposite. Phytates as salts of phytic acid bind not only P but also other elements that are not usable. For this purpose, it is advisable to use low phytate pea lines.

There is currently no standard for phytate content for animal and human nutrition. This parameter is variable depending on the growing conditions and especially on the genotype.  This genotypic screening is currently being evaluated and is part of another forthcoming publication in our authors' collective. The model pot experiment allowed to observe under defined growth conditions the factor of foliar application of P. We believe that the results of the experiment will serve to assess the possibility of practically available nutrient application (phosphorus) and to evaluate its effect on yield and grain quality (especially phytic acid content in pea seeds).

Reviewer's comment:

The results of Table 3 are discussed without referring to Table 3 in the text.

Authors' response:

Table 3 is referred to in the text (part 2.3. The Effect of Foliar P Fertilization on Pea Seed Quality) twice in the first reflection and once in the second reflection

Reviewer's comment:

The amount of phytates shown in Table 3 do not allow classifying the genotypes in “Normal” and “low” phytate.

Authors' response:

Pea genotypes were divided into two groups, the low-phytate pea lines (1-2347-144 and 1-150-81) and normal-phytate cultivars (CDC Bronco and Cultass) according to literature sources [50-53].

Warkentin, T.D.; Delgerjav, T.; Arganosa, G.; Rehman, A.U.; Bett, K.E.; Anbessa, Y.K; Rossnagel, B.; Raboy, V. Development and characterization of low-phytate pea. Crop Sci. 2012, 52, 74–78.

Warkentin, T.; Vandenberg, A.; Banniza, S.; Slinkard, A. CDC Bronco field pea. Can. J. Plant Sci. 2005, 85, 649–650.

Warkentin, T.D.; Delgerjav, O.; Arganosa, G.; Rehman, A.U.; Bett, K.E.; Anbessa, Y.; Rossnagel, B.; Raboy, V. Development and Characterization of Low-Phytate Pea. Crop. Sci. 2012, 52, 74–78.

Blade, S.; Warkentin T.; Vandenberg, A. Cutlass field pea. Can. J. Plant Sci. 2004, 84(2), 533-534.

Reviewer's comment:

-The effect of P foliar fertilization on phytate, protein, ash and mineral content of seeds is statistically significant exclusively for certain treatments for some of the genotypes. Moreover, in the few cases when the differences were significant, it is difficult to evaluate the physiological relevance of the effect. For example, with respect to K content, the only significant variation was for 1-2347-144 with P3, and the difference was from 0.92 to 1.01 %.

-In my opinion, the results show that P foliar fertilization increases seed yield of pea, with minor changes of seed quality.

Authors' response:

Our study does not confirm a consistent tendency towards the change of selected qualitative parameters (especially crude protein and mineral content in seed) under phosphorus fertilization of pea. The manuscript (abstract results and conclusions) was edited according to the recommendations.

CONCLUSIONS

Reviewer's comment:

Conclusions are a summary of results, without highlighting their significance.

Some of the conclusions are not reliable. The authors say that: “Responses of the low-phytate lines on the fertilization were more remarkable due to a significant increase of the seed production even at the highest P dose which reduced the seed production of the normal-phytate genotype”. This is only true when the effect of P2 is compared with P3·, but not when P3 is compared with P0 (Figure 1). With respect to control, P3 increased seed yield and seed number in all the genotypes. In addition, at P3 the seed yield and number was higher in normal-phytate genotypes.

Authors' response:

As stated in the Instructions for Authors, the section “conclusions” is not mandatory. The text summarizing the results has been removed, the last paragraph has been edited and inserted at the end of the section “Results and Discussion”.

Round 2

Reviewer 2 Report

The authors have responded to the reviewer's questions. However, the manuscript could be further improved by including some of the items discussed in the “response to reviewer” in the manuscript.

Author Response

We supplemented the manuscript with some of the items discussed in the “response to reviewer” from round 1 of the review. Thank you again for your constructive comments. We believe that taking them into account will improve the quality of the manuscript.
